# Self-Compassion in Irish Social Work Students: Relationships between Resilience, Engagement and Motivation

**DOI:** 10.3390/ijerph18158187

**Published:** 2021-08-02

**Authors:** Yasuhiro Kotera, Freya Tsuda-McCaie, Ann-Marie Edwards, Divya Bhandari, Geraldine Maughan

**Affiliations:** 1College of Health, Psychology and Social Care, University of Derby, Derby DE22 1GB, UK; freya.mccaie@googlemail.com (F.T.-M.); annm.edwards@icloud.com (A.-M.E.); 2Medical Governance Research Institute, Tokyo 1080074, Japan; rayordeal3@gmail.com; 3Department of Applied Social Sciences, Limerick Institute of Technology, V94 EC5T Limerick, Ireland; Geraldine.Maughan@lit.ie

**Keywords:** self-compassion, resilience, intrinsic motivation, engagement, Irish students, social work students

## Abstract

Self-compassion recognises a meaning of life’s suffering, aligning with existential positive psychology. Although this construct is known to protect our mental health, how to augment self-compassion remains to be evaluated. Social work students suffer from high rates of mental health problems; however, research into self-compassion in this population remains to be developed. This study aimed to evaluate (i) relationships between self-compassion and more traditional positive constructs—resilience, engagement and motivation, and (ii) differences of these constructs between the levels of studies to inform how self-compassion can be enhanced in social work students. A total of 129 Irish social work students completed self-report scales regarding self-compassion, resilience, engagement and motivation. Correlation, regression and one-way MANOVA were conducted. Self-compassion was associated with gender, age, resilience, engagement and intrinsic motivation. Resilience and intrinsic motivation were significant predictors of self-compassion. There was no significant difference in the levels of these constructs between the levels of studies. Findings suggest that social work educators across different levels can strengthen students’ resilience and intrinsic motivation to cultivate the students’ self-compassion. Moreover, the close relationships between self-compassion, resilience and intrinsic motivation indicate that orienting students to the meaning of the studies helps their mental health.

## 1. Introduction

### 1.1. Self-Compassion and Existential Positive Psychology

There is a significant commonality between self-compassion and Existential Positive Psychology (EPP). EPP is an integrative, holistic model, which conceptualises people as bio–psycho–social–spiritual beings and seeks to integrate the strengths and flourishing focused perspectives of positive psychology with the focus on meaninglessness, death and alienation prevalent in existential psychology [1]. Thus, EPP draws attention to the dialectical and interacting relationship between flourishing and suffering, conceptualising them as in opposition but at the same time existing in intimate relation with the other [2,3]. Additionally, EPP recognises that development and change can occur through the dynamic interplay of such opposites [2].

Self-compassion is having awareness of and bringing compassion to one’s own and others’ suffering and is conceptualised as encompassing three components: (i) self-kindness, (ii) common humanity and (iii) mindfulness [4]. Self-kindness is bringing an attitude of understanding and gentleness towards oneself, rather than harsh criticism, especially during times of hardship, pain or failure. Common humanity is the recognition that one’s suffering and failure are a part of the human condition and thus connect one with other people, rather than isolate one from them; that is, suffering and failure are a normal part of existence and therefore do not make one abnormal. Mindfulness is the practice of being aware of and acknowledging one’s difficult thoughts and feelings while not over-identifying them with the self [4].

Self-compassion is aligned with Existential Positive Psychology (EPP) in that it recognises the suffering inherent in living while also finding meaning and connection with others in that suffering. Self-compassion is a holistic approach in that it is concerned with both mitigating suffering or psychopathological symptoms and also with fostering flourishing and meaningful living. Indeed, research supports the connection of self-compassion to both reduced psychopathology and psychological flourishing [5]. Furthermore, in self-compassion, negative feelings or experiences are not rejected; rather, ‘positive emotions are generated by embracing the negative ones’ [5]. Thus, like EPP, self-compassion is an integrative, dialectical approach, as suffering and flourishing are seen as interacting: flourishing can occur through accepting and understanding suffering [6]. Lastly, mindfulness-based activities, which incorporate elements of self-compassion, have been used as a strategy to address death anxiety [7], a core concern of existential psychology.

Though still nascent, self-compassion has begun to be regarded as essential in social work for self-care and empathetic client care. Self-care was predicted by self-compassion among professional social workers [8], and self-care is understood as vital to prevent burnout and staff turnover [9]. Additionally, overidentification, which indicates a lack of self-compassion, was particularly strongly associated with depression among social work students in the United States [10]. Moreover, self-compassion fosters professional social workers’ empathy and commitment, enabling them to thrive in social work [11]. Professional social workers are generally self-compassionate; however, levels vary significantly according to a variety of factors, including health and education level [9,10]. These findings indicate a need to increase self-compassion among all social workers by understanding self-compassion and incorporating it in social work education [12].

### 1.2. Improving Self-Compassion: Interventions and Their Limitations

Understanding the relationship between self-compassion and other constructs is important in developing varied approaches for cultivating self-compassion, as self-compassion interventions remain to be further developed [13]. Existing interventions aimed at increasing levels of self-compassion, for example, the Mindfulness Self-Compassion programme [5], rely heavily on meditative techniques. Meditation may not be an optimal approach for all people [14]. Individuals with trauma histories may find meditating challenging as traumatic memories may arise during meditation [15]. Additionally, individuals prone to panic, feelings of emptiness or extreme dissociation may have negative experiences while practising mindfulness [16]. Social work appears to attract ‘wounded healers’ [17], quoted in Straussner, Stenrish and Steen, 2018, [18] (p. 1), and students frequently experience vicarious trauma [19,20]. Given that social work students may have memories or feelings that make meditative approaches challenging, identifying alternative interventions to cultivate self-compassion may be especially important. Identifying the relationship between self-compassion the other constructs explored here—motivation and engagement—may indicate such potential alternatives.

### 1.3. Self-Compassion and Mental Health and Wellbeing

Self-compassion is associated with increased wellbeing and decreased symptoms of poor mental health [21]. Having an understanding of one’s weaknesses can reduce self-criticism and shame, leading to better mental health [21]. Indeed, a meta-analysis synthesising evidence from 20 samples in 14 studies identified a large effect size between compassion and psychopathology [22]. Among social work students in the United States, high levels of self-compassion have been associated with lower levels of depressive symptoms [10], and in the UK, high levels of self-compassion were associated with better mental health [23]. Among university students more broadly, self-compassion has been associated with lower levels of distress and improved mental health [6,24,25,26]. Despite the importance of self-compassion for mental health, there is limited research exploring it in relation to other, more established constructs among social work students. Identifying a relationship between self-compassion and established constructs can offer insights into developing effective interventions to enhance self-compassion.

### 1.4. The Mental Health of Social Work Students

While poor mental health is common among UK university students generally [27], poor mental health is especially prevalent among social work students. Symptoms of depression impact 34–50% of this population [28,29,30]. Anxiety is also common [30], and 12% of social work students report suicidal ideation [28]. There are multiple possible contributors to the prevalence of mental health problems within this population. The caring professions appear to attract individuals with histories of hurt and/or childhood adversity, who may be more vulnerable to mental health issues [18,31]. Furthermore, the social work training programme is emotionally taxing and may contribute to mental health problems; for example, social work students reporting experiencing secondary traumatic stress [19,20] and emotional exhaustion [32]. Upon entering the profession, social workers are likely to experience work-based emotional distress [33], vicarious trauma [34] and burnout [35]. Lastly, the COVID-19 pandemic and its impacts on learning and social work may have further strained the mental health of social work students [36].

Poor mental health is associated with lower achievement and dropout among university students [37] and, more specifically, social work students’ academic success and course completion [38]. Thus, finding ways to improve mental health among social work students is important. An integrative approach to suffering and flourishing within the EPP framework may be especially relevant to social workers because of the likelihood of encountering such suffering through working with clients [39]. Research has begun to focus on the potential for post-traumatic growth—an inherently integrative, dialectical concept [2]—following vicarious healthcare workplace trauma, identifying the potential for positive change to occur through such trauma in certain conditions [40,41]. Similar to self-compassion [42], meaningfulness (e.g., religious engagement and enriching relationships) was associated with post-traumatic growth [43]. These findings suggest both self-compassion and EPP are especially helpful for the mental health of social work students.

### 1.5. Resilience

While there is no agreed-upon definition of resilience [44], the American Psychological Association (2014) defines resilience as ‘the process of adapting well in the face of adversity, trauma, tragedy, threats or even significant sources of stress’ [45]. Resilience has also been defined as the ability to establish healthy functioning following adverse events [46], while others stress the role of making an active decision to move forward from traumatic experiences—regardless of the presence of pathology—in defining resilience [44,47]. In the social work context, resilience has been described as remaining calm and empathetic with clients and demonstrating self-awareness, optimism and stability [48].

Cultivating resilience has been identified as an important goal for social work training. High levels of resilience are associated with lower levels of mental health problems among students [25] and decreased mental health problems and improved wellbeing in the workforce [49]. Additionally, resilience has been identified as potentially protective against the negative impacts of work-related stress [50]. Given the stressful nature of social work, integrating resilience training into the curriculum has been identified as a vital part of education and training [51]. In the workforce, resilience training has been effective in improving resilience with consequent positive impacts on mental health and wellbeing [49]. Among social workers, interventions designed to improve resilience have demonstrated efficacy in reducing perceived work-related stress and fostering positive attitudes, perspectives and behaviours [52]. Self-care and resilience-building approaches are integrated into the curriculum in some Irish social work education programmes, and students must meet self-care standards as part of demonstrating fitness to practice [53]. Thus, understanding the relationship between resilience and self-compassion may suggest fruitful avenues for integrating self-compassion training into existing social work training.

### 1.6. Motivation

Self-Determination Theory (SDT), aligning with EPP’s concerns, articulates the roles of autonomy and authenticity in motivating human behaviour and their potential impact on flourishing and psychological wellbeing [54]. SDT posits that three innate psychological needs motivate human behaviour: autonomy, competence and relatedness, and it distinguishes between two forms of motivation: extrinsic and intrinsic [55]. Intrinsic motivation describes motivation to participate in an activity because it is inherently satisfying and interesting to do so, in contrast to extrinsic motivation, in which behaviour is motivated by a desire to achieve a certain end (for example, money or a high grade). Intrinsic motivation is associated with numerous positive impacts: mental wellbeing [56], performance quality [57], determination [58], job satisfaction [59], reduced burnout [3] and life satisfaction [60]. In university settings, intrinsic motivation is associated with improved self-esteem [61] and academic outcomes [62], while among social work students, it is associated with perceived skill and satisfaction with learning experiences [63]. Extrinsic motivation is associated with mental health problems [59], shame [64], workplace and academic burnout [65,66]. However, extrinsic motivation in combination with intrinsic motivation is associated with improved performance [57]. Despite the relationship between intrinsic motivation and positive outcomes, the impact of intrinsic motivation on self-compassion is limited, suggesting a need for evaluation.

### 1.7. Engagement

Engagement is another construct relevant to mental health. Engagement describes the effort and determination students exhibit in learning or mastering skills and in acquiring knowledge [67]. Schaufeli et al. [68] argue that engagement is comprised of three components: vigour, dedication and absorption. Vigour describes investing high levels of energy, effort and persistence in learning. Dedication describes feeling enthusiastic about and proud of one’s work. The feeling that time is passing swiftly and being immersed in one’s studies characterises absorption [68]. Engagement is related to improved academic performance and successful completion of a degree [67,69]. Engagement is also associated with positive impacts beyond the academic context, such as improved mental health [70] and increased resilience and reduced burnout [71], while being negatively associated with compassion fatigue [72]. Among social work students, engagement was negatively related to mental health problems [23]. The relationships between self-compassion, resilience, engagement and motivation have not been explored with Irish social work students. Moreover, how those constructs differ by the levels of studies remains to be appraised.

### 1.8. Aims

Accordingly, this study aims to evaluate relationships between self-compassion, resilience, engagement and motivation (Aim 1), identify significant predictors of self-compassion (Aim 2), and appraise differences in the levels of these psychological constructs between different levels of studies (Aim 3). Aims 1 and 2 were designed to identify which of these more established constructs were associated with this relatively new construct, self-compassion. Aims 3 was designed to provide insights into whether the current curriculum may significantly impact self-compassion and the other variables across different levels.

## 2. Materials and Methods

### 2.1. Participants

Participants were 18 years old or older and enrolled in a social work programme at two universities in the Republic of Ireland. An online survey was disseminated to the students via module announcement by their tutors (authors were not involved); therefore, students who were on a study break were excluded. Based on convenience sampling, 140 students in total were approached, of which 129 undergraduate students (109 females (84%), 16 males (12%), 4 did not answer (3%); Age M = 25.12, SD = 7.68, Range = 18–47 years old; 68 first-year, 40 third-year and 21 fourth-year students (second-year students were not approached as they were introduced to other research)) agreed to take part (92% response rate), and completed self-report scales regarding self-compassion, resilience, engagement and motivation. The required sample size per power analysis was satisfied (84: two tails, *p* H1 = 0.30, α = 0.05, Power = 0.80, *p* H0 = 0) [73]. The gender balance and age in our sample were similar to the general social work student demography (86% female, Age M = 28 years old, combining both undergraduates and postgraduates; Skills for Care, 2016). No compensation was awarded for participation. Following the ethical guidelines, the withdrawn 16 students were not asked for the reason; no reason nor complaint was received. Ethical approval was granted by the Ethics Committee of the Limerick Institute of Technology on 1 December 2018. The same participants were also included in a parallel study by the authors [74].

### 2.2. Materials

Four scales were used regarding self-compassion, resilience, engagement and motivation.

Self-compassion was measured using the Self-Compassion Scale-Short Form (SCS-SF), a shortened version of the full 26-item Self-Compassion Scale [4]. SCS-SF consists of 12 items responded on the five-point Likert scale (e.g., ‘When something upsets me, I try to keep my emotions in balance’; 1 = ‘Almost never’ to 5 = ‘Almost always’; inverted scores are used for items 1, 4, 8, 9, 11 and 12). Cronbach’s alpha was high (0.86) [75].

Resilience was assessed using the 6-item Brief Resilience Scale (BRS; Smith et al., 2008) [76] was used to measure the level of resilience, the ability to bounce back from adversity [77]. The six items (e.g., ‘It does not take me long to recover from a stressful event.’) are responded on the five-point Likert scale (1 = ‘Strongly disagree’ to 5 = ‘Strongly agree’; inverted scores for the items 2, 4 and 6). BRS had high internal consistency (α = 0.80–0.91; [77]).

The 17-item Utrecht Work Engagement Scale for Students (UWES-S) appraised the level of engagement [78]. The 17 items capture three aspects of engagement: (a) vigour refers to the energy that leads to effort in studies (six items, e.g., ‘When I get up in the morning I feel like going to class’); (b) dedication refers to commitment to studies (five items, e.g., ‘I am proud of my studies’); and (c) absorption (positive immersion in academic work; six items, e.g., ‘Time flies when I am studying’). A seven-point Likert scale was used for responses (0 = ‘Never’ to 6 = ‘Always (everyday)’) [78]. High internal consistency was identified in UWES-S (α = 0.63–0.81) [78]. For the purposes of this study, the average of the global score for the engagement measure was used (α = 0.93; [78]).

Lastly, motivation was assessed using the Academic Motivation Scale [79], which asks students why they go to a university. AMS comprises 28 items considering three types of motivation: (a) intrinsic motivation relates to inherently pleasure from engaging in the activity (12 items, e.g., ‘Because I experience pleasure and satisfaction while learning new things’); (b) extrinsic motivation relates to engaging in the activity as a means to an end (12 items, e.g., ‘In order to obtain a more prestigious job later on’); and (c) amotivation relates to no motivation to study at all (four items, e.g., ‘I don’t know; I can’t understand what I am doing in school’). Items are responded on a seven-point Likert scale (1 = ‘Does not correspond at all’ to 7 = ‘Corresponds exactly’). AMS demonstrated adequate to high internal consistency (α = 0.62–0.91; [79]).

### 2.3. Procedure

First, data screening was performed to detect any outliers and check the assumptions of parametric tests. Second, correlation and regression analyses were performed to identify correlates with and predictors for self-compassion. Finally, one-way MANOVA was conducted to appraise whether the levels of self-compassion, resilience, engagement and motivation were different by their levels of studies. As self-compassion, resilience, extrinsic motivation and amotivation were not normally distributed (Shapiro-Wilk’s test, *p* < 0.05), square-root-transformation was used to satisfy the assumption of normality [80].

## 3. Results

Analyses were conducted using IBM SPSS version 26.0. No outliers were identified. All variables demonstrated good internal reliability in our sample (α = 0.76–0.94; Table 1).

### 3.1. Relationships with Self-Compassion (Aim 1)

Pearson’s correlation was calculated (Table 2). Gender and age were included to inform our following analysis using a regression model. Self-compassion was positively associated with gender, age, resilience, engagement and intrinsic motivation, whereas it was not associated with extrinsic motivation and amotivation.

### 3.2. Predictors of Self-Compassion (Aim 2)

To identify significant predictors of self-compassion from significant correlates, namely resilience, engagement and intrinsic motivation, multiple regression analyses were performed (Table 3). At step one, gender and age were entered to adjust for their effects. At step two, resilience, engagement and intrinsic motivation were entered as predictor variables. Multicollinearity was of no concern (VIF < 10). The adjusted coefficient of determination (Adj. *R*^2^) was reported. Resilience, engagement and intrinsic motivation yielded 14% of the variance for self-compassion, indicating a medium effect size [40]. Resilience and intrinsic motivation were significant predictors of self-compassion, where resilience (β = 0.36) predicted self-compassion more strongly than intrinsic motivation (β = 0.25). Further relative contribution analysis [81] confirmed the strength of each variable (resilience 12%, intrinsic motivation 2% and engagement < 1%).

### 3.3. Differences by Level of Studies (Aim 3)

A one-way MANOVA was run to determine the effect of the students’ level of studies on self-compassion, resilience, engagement and motivation (Table 4). There was homogeneity of variance-covariances matrices, as assessed by Box’s test of equality of covariance matrices (*p* = 0.058). There was not a statistically significant difference between the levels on the combined dependent variables, F (12, 242) = 1.18, *p* = 0.30; Wilks’ Λ = 0.89; partial η^2^ = 0.06.

## 4. Discussion

There is little research in the literature on the mental health of social work students, despite high academic expectations being a major cause of stress in this particular group [82]. We investigated self-compassion in a sample of Irish social work students. Additionally, we were also interested in examining the relationships between the more traditional positive constructs used in this research, how these variables differ between the levels of studies, and how self-compassion can be enhanced in social work students. Regarding Aim 1, self-compassion was positively associated with gender, age, resilience, engagement and intrinsic motivation. Moreover, resilience and intrinsic motivation were significant predictors of self-compassion, with resilience being the strongest predictor (Aim 2). Lastly, there was no significant difference reported in the levels of these constructs between the levels of studies (Aim 3). Self-compassion was not associated with extrinsic motivation and amotivation. These findings are discussed in detail below.

In the present study, a positive relationship between self-compassion, gender, age, resilience, engagement and intrinsic motivation was noted, consistent with previous findings showing the positive impact of self-compassion in academic settings [25,83,84,85]. Students who cultivate self-compassion are likely to be more motivated [86], resilient and engaged [83]. Moreover, self-compassion is directly related to academic performance [87] and mental wellbeing [82]. While motivation is important for academic success, students may struggle to recover from academic setbacks if they lack resilience (fear of failure, poor performance, academic pressure and negative feedback) [88]. Accordingly, prior research indicates that high levels of resilience protect against the negative impacts of stress-related illnesses and provides improved learning and academic achievement [6,89]. Moreover, the literature links academic resilience and engagement, positively influencing personal and academic performance [90]. The relationship between the positive psychological constructs of motivation, engagement, resilience and self-compassion is key to achieving academic success. Moreover, we identified that resilience and intrinsic motivation were significant predictors of self-compassion, echoing previous findings: students who are able to bounce back from difficulties (i.e., resilience) and are passionate about their studies (i.e., intrinsic motivation) tend to have higher levels of kindness towards themselves [83,91,92]. In a recent study by Edgar et al. [93], student success is influenced by self-belief, which has been identified as a key dimension of motivation. Future work may be useful to investigate particular aspects of self-belief because research indicates it has a significant influence on motivation and resilience [94], which can impact self-compassion.

Contextualising our findings in social work, indeed, social work does require a high level of self-compassion for practitioners to maintain wellbeing and performance [11]. Social workers themselves are often wounded healers [17] and encounter emotionally challenging situations [15]. While offering care for their service users, they need to take care of themselves [95]. Self-compassion is essential for good self-care among social workers [8]. Our findings can offer insights that their current emphasis on resilience in the professional framework [48] is somewhat conducive for cultivating self-compassion among social workers. Considering the ever-increasing importance of social workers [96], interventions and evaluation to enhance self-compassion need to be implemented. Our findings offer the foundational data to support such new practice and research.

The study found that the level of studies on self-compassion was not significantly different; that is, the year of the study did not appear to affect levels of self-compassion among participating students. Conflicting results have been reported in the literature by Gunnell et al. [97], who suggested that a change in self-compassion was linked to a change in wellbeing and emphasised the importance of students cultivating self-compassion during their first year at university to help mitigate a decline in wellbeing over time. Further longitudinal studies in students at different time points and from different degrees could explore this further.

Lastly, the results of our study can inform the field of EPP. Intrinsic motivation and resilience were particularly strongly associated with self-compassion. These constructs share qualities of EPP. Intrinsically motivated students find meaning in their studies and enjoy their learning, as opposed to extrinsically motivated students who recognise the value of studying in an external reward they are after (e.g., grades). This, somewhat, materialistic motivation was not associated with self-compassion, whereas more meaning-oriented intrinsic motivation was. Likewise, resilience refers to the adaptable process of dealing with stress in a healthy way by employing positive coping mechanisms [90,98,99]. Negative experiences have a lower impact on students who are more resilient [100]. Positive emotions are thought to improve problem-solving abilities and have a direct effect on psychological wellbeing [101], so students who are more resilient tend to experience fewer mental health issues [99] and will likely achieve greater academic success [90].

## 5. Limitations

Although this study offers useful insights into the self-compassion of social work students, some limitations of the current study should be noted. First, because correlation and regression analyses were employed in this study, causality could not be determined. A longitudinal design could prove useful in future studies. In addition, the study relied on self-reported measures, which restricts the reliability of participant responses and may influence results [102]. Future research should employ experimental designs that will provide additional insight into the role of self-compassion in academic settings. Further, the Self-Compassion Scale—Short Form was used to measure self-compassion; however, its accuracy has yet to be proven [103]. In addition, because most of the participants were women, the findings may not generalise to men. Relatedly, due to an ethical consideration that some students might feel uncomfortable disclosing their demographic data in a small sample, we did not ask which of the two institutions they belonged to, thus did not evaluate the institutional differences of the variables between the two institutions, which could inform our analyses. Lastly, since all participants were social work students, the study conclusions cannot be generalised to other degrees. Further research is warranted in this area.

## 6. Conclusions

Though the importance of self-compassion to the mental health of social work students has been reported, how this relatively new construct could be cultivated remained to be evaluated among Irish social work students. Our results indicated that resilience and intrinsic motivation—more traditional constructs—were strongly associated with self-compassion, suggesting effective means to improve self-compassion. The meaning orientation of both resilience (e.g., coping with adversities flexibly) and intrinsic motivation (e.g., paying attention to intrinsic experience) may mean that EPP approaches can be helpful to enhance self-compassion, leading to better mental health. For example, educating students to identify and embrace the meaning of their studies may be helpful to protect their mental health. Future research is warranted to implement and evaluate the effects of EPP interventions to improve self-compassion among Irish social work students.

## Figures and Tables

**Table 1 ijerph-18-08187-t001:** Descriptive statistics: self-compassion, resilience, engagement and motivation in Irish social work students (*n* = 129).

Scale	Variable (Range)	M	SD	α
	Age (18–47)	24.70	7.22	
Self-Compassion Scale-Short Form	Self-Compassion (1–5)	2.81	0.58	0.79
Brief Resilience Scale	Resilience (1–5)	3.00	0.71	0.76
Utrecht Work Engagement Scale for Students	Engagement (0–6)	3.20	1.16	0.94
Academic Motivation Scale	Intrinsic Motivation (1–7)	4.73	1.23	0.93
	Extrinsic Motivation (1–7)	5.44	1.06	0.90
	Amotivation (1–7)	2.44	1.55	0.87

**Table 2 ijerph-18-08187-t002:** Correlations among self-compassion, resilience, engagement and motivation in Irish social work students (*n* = 129).

No.	Variables	1	2	3	4	5	6	7	8
1	Self-Compassion	-	0.37 **	0.18 *	0.26 **	0.05	−0.03	0.18 *	0.20 *
2	Resilience		-	0.11	0.02	0.11	−0.02	0.15	0.04
3	Engagement			-	0.67 **	0.40 **	−0.43 **	0.16	0.35 **
4	Intrinsic Motivation				-	0.52 **	−0.35 **	0.19 *	0.31 **
5	Extrinsic Motivation					-	−0.25 **	0.14	0.10
6	Amotivation						-	0.19 *	−0.23 **
7	Gender (0 = F, 1 = M)							-	0.05
8	Age								-

* *p* < 0.05, ** *p* < 0.01.

**Table 3 ijerph-18-08187-t003:** Multiple regression: resilience, engagement and intrinsic motivation to self-compassion among social work students (*n* = 129).

Outcome Variable: Self-Compassion	B	SE_B_	β	95% Confidence Interval for B	
				Lower Bound	Upper Bound
Step 1					
Gender (0 = F, 1 = M)	0.09	0.05	0.17	−0.002	0.19
Age	0.01	0.002	0.19 *	0.001	0.01
Adj. *R*^2^		5%			
Step 2					
Gender (0 = F, 1 = M)	0.04	0.05	0.08	−0.05	0.13
Age	0.003	0.002	0.13	−0.001	0.01
Resilience	0.32	0.07	0.36 **	0.18	0.46
Engagement	−0.04	0.05	−0.09	−0.15	0.06
Intrinsic Motivation	0.16	0.07	0.25 *	0.02	0.29
ΔAdj. *R*^2^		14%			

B = unstandardised regression coefficient; SE_B_ = standard error of the coefficient; β = standardised coefficient; Adj. *R*^2^ = adjusted coefficient of determination; ΔAdj. *R*^2^ = delta adjusted coefficient of determination; * *p* < 0.05, ** *p* < 0.01.

**Table 4 ijerph-18-08187-t004:** One-way MANOVA: difference in the mean self-compassion, resilience, engagement and motivation between different levels of studies.

Level of Studies (*n*)	Self-Compassion		Resilience		Engagement		Intrinsic Motivation		Extrinsic Motivation		Amotivation	
	M	SD	M	SD	M	SD	M	SD	M	SD	M	SD
1st Year (68)	2.79	0.61	3.05	0.84	3.30	1.17	4.71	1.18	5.47	0.96	2.33	1.55
3rd Year (40)	2.90	0.51	2.88	0.48	3.11	1.18	4.82	1.15	5.21	1.17	2.68	1.60
4th Year (21)	2.70	0.61	3.05	0.57	3.08	1.13	4.62	1.54	5.77	1.10	2.35	1.47
Total (129)	2.81	0.58	3.00	0.71	3.20	1.16	4.73	1.23	5.44	1.06	2.44	1.55

No significant difference was found.

## Data Availability

The data presented in this study are available on request from the corresponding author. The data are not publicly available due to the privacy of research participants.

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
