# Peer review of "Self-Compassion in Irish Social Work Students: Relationships between Resilience, Engagement and Motivation"

_ijerph, 2021, doi:10.3390/ijerph18158187_

Round 1

Reviewer 1 Report

Thank you for the opportunity to review the manuscript “Self-Compassion in Irish Social Work Students: Relationships between Resilience, Engagement and Motivation” (ijerph-1321535).

The manuscript deals with relevant constructs in psychological and educational contexts. It has the potential to add to the current research literature.

My main comments are presented in the attached archive.

Best regards

Author Response

Thank you for your helpful feedback. We have systematically revised our manuscript addressing the points you have raised. Please see our responses below. We hope this revised paper is now acceptable for publication. We extend our sincere gratitude to you for your feedback that has significantly helped to strengthen the paper.

Reviewer 1

Reviewer 1’s comment 1

perhaps too many self-citations are included.

Authors’ response 1-1

In line with your comment, now self-citations are reduced. We did not want to break the format, so those changes are marked using comments.

Reviewer 1’s comment 2

mental health has not been evaluated, therefore these statements should be hypotheses

Authors’ response 1-2

In line with your comment, now ‘mental health’ in L23 is removed. We would like to keep the one in L25, because mental health is introduced in this study as that is what self-compassion contributes to (L11). However, we believe that removal of the one in L23 will give clarity and avoid possible confusion for readers.

Reviewer 1’s comment 3

Engagement?

Authors’ response 1-3

In line with your comment, now engagement is added to keywords.

Reviewer 1’s comment 4

It would be interesting to describe the main characteristics of social work studies and the tasks / skills of these professionals. Why is self-compassion important for social workers?

Authors’ response 1-4

Thank you for your helpful suggestion. In line with your comment, now a new paragraph is added, describing the relationship between social work and self-compassion.

Reviewer 1’s comment 5

This section should be expanded (for example, providing more information from [8])

You can also consult this article:

https://doi.org/10.1016/j.cpr.2012.06.003

Authors’ response 1-5

Thank you for your helpful suggestion. In line with your comment, more details are provided from the already-cited study and your suggested study.

Reviewer 1’s comment 6

this section should be moved to the end of the introduction, just before Aims

Authors’ response 1-6

In line with your comment, now it is moved to before Aims. Thank you.

Reviewer 1’s comment 7

these words can be omitted to leave more space in the M, SD and alpha columns

Authors’ response 1-7

Table is now adjusted as suggested. Thank you.

Reviewer 1’s comment 8

Taking into account the characteristics of the sample, I think that the variable sex / gender should not be evaluated

Authors’ response 1-8

Thank you for your suggestion. However, findings here (i.e., significant correlations with self-compassion) contribute to a more accurate regression model, where we adjusted for those two variables, so we would like to keep them. Clarifying statements are now added.

Reviewer 1’s comment 9

Considering that self-compassion is the main variable, I think it should be the first one in the table.

Authors’ response 1-9

In line with your comment, now the table is adjusted.

Reviewer 1’s comment 10

conclusions should be deeper

Authors’ response 1-10

In line with your comment, now the conclusion section has gained some more depth, while maintaining the conciseness as instructed in the journal’s guidelines.

https://www.mdpi.com/journal/ijerph/instructions#manuscript

Other issues:

Missing references indicated in red are now inserted. Thank you.

Reviewer 2 Report

This is a well written paper, and the analyses are appropriate for the data collected. However, I think the authors need to develop further the justification for conducting and reporting the study.

Structure of introduction. By and large the introduction is well written, but it is highly descriptive and does not problematise the issues sufficiently. Rather, the introduction starts with a description of concepts and inter-relationships, rather than establishing the need for the current study through a critical analysis of gaps in the literature. In the same vein, the research questions are not really introduced, explained or justified (especially differences between levels of studies).

Methods. Are there differences between the samples from the two institutions and in response rates? 92% is a very high response rate. How was this achieved?

Engagement scale – could you provide the overall alpha and reason for aggregation please?

Analyses. If there are differences on the key variables between institutions, then this needs to be controlled in the analyses. This could be especially problematic for the MANOVA if there are differences in the proportion of students from different levels of study.

P 6 249-253. Some form of dominance analysis would be needed to justify the claim one predictor is stronger than another would it not?

P 7. Second and third paragraphs of the discussion seem to repeat very similar material and make very similar points – suggest integrate the two into a single paragraph. In relation to the comment about the introduction, this discussion may need to be restructured to describe how the study addresses the significant gaps identified in the introduction.

Minor

P 2, 54-55 ‘positive emotions are generated by embracing [author’s own] the negative ones’ – author’s own is misdirecting here, so meaning of sentence is unclear

P 5 199-200 ‘Resilience was assessed using the 6-item Brief Resilience Scale (BRS; Smith et al., 2008) was used to measure the level of resilience, the ability to bounce back from 200 adversity [71].’ Seems to be both a typo and formatting issue here

Author Response

Thank you for your helpful feedback. We have systematically revised our manuscript addressing the points you have raised. Please see our responses below. We hope this revised paper is now acceptable for publication. We extend our sincere gratitude to you for your feedback that has significantly helped to strengthen the paper.

Reviewer 2

Reviewer 2’s comment 1

This is a well written paper, and the analyses are appropriate for the data collected. However, I think the authors need to develop further the justification for conducting and reporting the study.

Authors’ response 2-1

Thank you for your kind words. In line with your comment, links between self-compassion and social work have been more highlighted (above 1.2) and the justification is now adjusted and placed before the study aims.

Reviewer 2’s comment 2

Structure of introduction. By and large the introduction is well written, but it is highly descriptive and does not problematise the issues sufficiently. Rather, the introduction starts with a description of concepts and inter-relationships, rather than establishing the need for the current study through a critical analysis of gaps in the literature. In the same vein, the research questions are not really introduced, explained or justified (especially differences between levels of studies).

Authors’ response 2-2

Thank you for your helpful suggestion. In addition to the extra paragraph about links between self-compassion and social work, the paragraph about aims is now revised to clarify why those research questions were established, linking with the literature review done in the introduction.

Reviewer 2’s comment 3

Methods. Are there differences between the samples from the two institutions and in response rates? 92% is a very high response rate. How was this achieved?

Authors’ response 2-3

We did not collect data about the institution because we were aware that a sample from each institution would not be large. From our previous experience with UK social work students, we have learned that some students feel uncomfortable reporting their demographic information if they know they are a minority in a small sample (e.g., sexual orientation). We believe that robust protection of participants such as this has contributed to the high response rate.

Reviewer 2’s comment 4

Engagement scale – could you provide the overall alpha and reason for aggregation please?

Authors’ response 2-4

No problem. Now inserted: Preliminary manual of the scale.

Reviewer 2’s comment 5

Analyses. If there are differences on the key variables between institutions, then this needs to be controlled in the analyses. This could be especially problematic for the MANOVA if there are differences in the proportion of students from different levels of study.

Authors’ response 2-5

Thank you for your insight. Due to the reason given at 2-3, we did not collect data about the institutional differences. Now this is added to the limitation section.

Reviewer 2’s comment 6

P 6 249-253. Some form of dominance analysis would be needed to justify the claim one predictor is stronger than another would it not?

Authors’ response 2-6

Thank you for your suggestion. As you noted, relative importance analysis such as dominance analysis needs to be reported. Given the limitation of SPSS, we conducted a model comparison approach, where we created two models (one includes the measuring variable and the other does not) to identify each importance. The results did not change, however this is now noted in the methods.

Reviewer 2’s comment 7

P 7. Second and third paragraphs of the discussion seem to repeat very similar material and make very similar points – suggest integrate the two into a single paragraph. In relation to the comment about the introduction, this discussion may need to be restructured to describe how the study addresses the significant gaps identified in the introduction.

Authors’ response 2-7

In line with your comment, now those paragraphs are integrated, and how our findings can address the gaps identified in the introduction are added.

 Reviewer 2’s comment 8

Minor

P 2, 54-55 ‘positive emotions are generated by embracing [author’s own] the negative ones’ – author’s own is misdirecting here, so meaning of sentence is unclear

P 5 199-200 ‘Resilience was assessed using the 6-item Brief Resilience Scale (BRS; Smith et al., 2008) was used to measure the level of resilience, the ability to bounce back from 200 adversity [71].’ Seems to be both a typo and formatting issue here.

Authors’ response 2-8

Thank you for spotting these. Now those parts are corrected.

Round 2

Author Response

Thank you for your helpful feedback. Now the adjustments are made in the results section, and additional clarification was made in the discussion section.

Reviewer 2 Report

Thank you for attending to my earlier comments. I still have some concerns about the structure of the introduction and making the case for the paper. Specifically, section 1.7. does more to establish the need for the study, but it needs more development and placed earlier in the paper to help the reader understand why the paper is worth reading. The rest of the introduction then needs to be redrafted to reflect the restructuring.

More specific points are:

4th para, section 1.1. 'These findings indicate a need to incorporate self-compassion in social work education [11].' But the study does not do this - this is more of a conclusion.

2nd para discussion 'The results suggest that intrinsically motivated students are more likely to have high levels of psychological resilience'. The results do not show this - there is not a significant correlation between resilience and either facet of motivation, so the new text needs to be changed to reflect this.

Limitations 'Relatedly, we did not evaluate the institutional differences of the variables between the two institutions' - why is this important - see my comments on the first version.

Author Response

Dear Reviewer 2,

Thank you for your helpful feedback again. We have systematically revised our manuscript addressing the points you have raised. Please see our responses below. We hope this revised paper is now acceptable for publication. We extend our sincere gratitude to you for your feedback that has significantly helped to strengthen the paper.

Reviewer 2

Reviewer 2’s comment 1

Thank you for attending to my earlier comments. I still have some concerns about the structure of the introduction and making the case for the paper. Specifically, section 1.7. does more to establish the need for the study, but it needs more development and placed earlier in the paper to help the reader understand why the paper is worth reading. The rest of the introduction then needs to be redrafted to reflect the restructuring.

Authors’ response 2-1

Thank you for your helpful feedback. In line with your comment, the section 1.7 has been placed earlier, and the rest of the introduction has been redrafted, highlighting the rationale for this study.

Reviewer 2’s comment 2

More specific points are:

4th para, section 1.1. 'These findings indicate a need to incorporate self-compassion in social work education [11].' But the study does not do this - this is more of a conclusion.

2nd para discussion 'The results suggest that intrinsically motivated students are more likely to have high levels of psychological resilience'. The results do not show this - there is not a significant correlation between resilience and either facet of motivation, so the new text needs to be changed to reflect this.

Limitations 'Relatedly, we did not evaluate the institutional differences of the variables between the two institutions' - why is this important - see my comments  on the first version.

Authors’ response 2-2

Thank you for your helpful feedback. The sentence in the 4th paragraph is now corrected, stating that these findings indicate a need to ‘understand’ self-compassion.

In line with your suggestion, the sentence in the 4th paragraph is now adjusted, and the need to incorporate self-compassion in social education is now discussed in the conclusion.

Thank you for spotting. Now the sentence in the 2nd paragraph in the discussion is removed because that is not directly related to the findings about prediction of self-compassion. Instead, the previous sentence is expanded to clarify the findings. Additionally, the last sentence in this paragraph is adjusted to refer to our findings. 

The sentence in the limitation about the institutional differences is now expanded to clarify why we did not evaluate them, and what they could inform about our analyses as your first comment noted.

Analyses. If there are differences on the key variables between institutions, then this needs to be controlled in the analyses. This could be especially problematic for the MANOVA if there are differences in the proportion of students from different levels of study.